# India: The Last and Best Frontier for Cystic Fibrosis Newborn Screening with Perspectives on Special Challenges

**DOI:** 10.3390/ijns11020027

**Published:** 2025-04-17

**Authors:** Philip M. Farrell, Grace R. Paul, Sneha D. Varkki

**Affiliations:** 1Departments of Pediatrics and Population Health Sciences, University of Wisconsin School of Medicine and Public Health, 600 Highland Avenue, Madison, WI 53792, USA; 2Division of Pulmonary and Sleep Medicine, Nationwide Children’s Hospital, 700 Children’s Drive, Columbus, OH 43025, USA; grace.paul@nationwidechildrens.org; 3Department of Paediatrics, Christian Medical College, IDA Scudder Road, Vellore 632004, Tamil Nadu, India; sneha@cmcvellore.ac.in

**Keywords:** cystic fibrosis, newborn screening, incidence, India, Indian subcontinent, cystic fibrosis transmembrane regulator gene, immunoreactive trypsinogen

## Abstract

Because a delayed diagnosis of cystic fibrosis (CF) is detrimental and may be fatal, screening at birth has become routine in the Western world and has proven beneficial for many reasons, in addition to enabling prompt specialized care. Newborn screening (NBS) programs have elucidated the true incidence of CF in a variety of populations and enabled rapid genotype identification through the analysis of the cystic fibrosis transmembrane regulator (*CFTR*) gene. NBS studies also have revealed regional and population differences in *CFTR* variants and refuted the dogma that CF is a “white person’s disease”. But some regions have not yet implemented CF NBS, particularly in Asia where the disease prevalence has been uncertain. While the needs of a few low-and-middle-income countries are being addressed sequentially, one of the regions of greatest current interest is the Indian subcontinent because of recent data suggesting a higher incidence than that previously assumed, and clinical observations indicating tragic outcomes due to delayed diagnoses or failure to diagnose the disorder in young children. Thus, we conclude that the opportunities for research combined with service in the Indian subcontinent are urgent and potentially very impactful. Consequently, India is the *last and best frontier* for CF NBS, as we argue herein.

## 1. Introduction—The Goals and Value of Newborn Screening

Newborn screening (NBS) has been defined [1] as a “population-based public health program applying preventive medicine in defined regions to reduce infant morbidity and mortality from certain biochemical and genetic disorders by using presymptomatic detection/diagnosis with dried blood specimens from newborns analyzed in central laboratories employing automated procedures linked to clinical follow-up systems”. Generally regarded as a public health program, the NBS system is in fact a hybrid requiring collaboration between laboratories and pediatric care facilities with “clinical follow-up systems” to achieve its goals efficiently and effectively. This partnership is paramount. Because early diagnosis must be inextricably linked to early treatment, an important impact has been catalyzing the creation of subspecialty follow-up centers with multidisciplinary teams—usually developed incrementally to cover designated geographic regions. These centers have made unique contributions through research because many discoveries about relatively uncommon hereditary metabolic disorders have depended upon observations and interventions before secondary disease consequences occur. In the molecular era, the value of research linked to NBS-facilitated early diagnosis has been extraordinary.

Although reducing infant mortality, as the often-quoted statement “newborn screening saves lives” proudly proclaims, is one of the goals, the usual benefit of early diagnosis through screening coupled to early treatment is a reduction in morbidity such as preventing brain damage in phenylketonuria (PKU). Thus, NBS per se has been a critical advance in promoting improved care through preventive medicine strategies.

## 2. Cystic Fibrosis Newborn Screening—Overcoming Skepticism Through Research

### 2.1. The Challenge of Demonstrating That the Benefits Outweigh the Risks of CF NBS

In the case of the Western world‘s relatively common autosomal recessive disorder, cystic fibrosis (CF), it was challenging to demonstrate the significant benefits of NBS after a good screening biomarker was discovered [2], namely, blood immunoreactive trypsinogen (IRT). In fact, the first two CF NBS programs in the USA developed in Colorado [3] and Wisconsin [4] were driven by research objectives focused on potential benefits. Data from these pioneering programs led the Centers for Disease Control and Prevention (CDC) and the U.S. Cystic Fibrosis Foundation (CFF) to recommend universal CF NBS in 2004 [5,6]. But it required the combination of a randomized clinical trial [7] to discover the significant nutritional advantages of early diagnosis/treatment, and an exceptionally large retrospective cohort (N = 9571) to show the improvement in lung function at 9 years of age [8].

Research has been especially important in CF to refute misunderstandings and even embedded but erroneous dogma. The following discoveries illustrate the value of NBS-associated research. The long-held belief in a 1:2000 CF incidence in the USA was quickly refuted when many lines of NBS data revealed it was actually about 1:4000 [4]. In fact, we learned that the incidence of CF can only be determined reliably through NBS. In addition, it was shown early in NBS studies that the use of 60 mmol/L as the sweat chloride threshold value for a CF diagnosis was improper because countless infants were identified with two pathogenic *CFTR* variants and symptoms but with chloride values in the 30–59 mmol/L range. Lastly, and perhaps the best example, we have the observations dispelling the notion that CF is a “white person’s disease” which the McColley team proved erroneous by analyzing 9 years of data in the CFF Patient Registry [9].

Yet, skepticism about the value of CF NBS and overt resistance to its implementation were dominant features of the CF clinical culture for three decades, beginning in the 1970s and peaking during the 1980s [10]. In retrospect, a surprising number of arguments against early diagnosis prevailed among influential leaders and delayed widespread implementation for many years. Fortunately, attitudes began to change after 1989 when the *CFTR* gene was discovered [11], and, soon thereafter, the IRT/DNA screening algorithm [4,7] followed, and molecular therapies became a realistic dream. In addition, the prior organization of multidisciplinary CF centers in many regions and even in entire large countries like the USA proved invaluable in achieving early follow-up care.

### 2.2. Internationalization of CF NBS

The rationale for universal CF NBS became compelling when the disorders’ detrimental impact, earlier than previously apparent, became widely recognized, concurrent with CFTR modulator therapy appearing on the horizon [12,13]. Attitudes changed and advocacy efforts became more effective during the first decade of the 21st century. Generally, implementation within each country was accomplished region-by-region rather than by national fiat. Some countries like Canada experienced extensive delays of 10 or more years after the first program began until nationwide screening was accomplished. But, as Scotet et al. [14] reported, CF NBS programs proliferated internationally during 2005–2020.

### 2.3. Remaining Global Needs and Challenges

Although benefits attributable to NBS for children with CF are routinely accomplished now in the Western civilization regions of North America, most of Europe, Australia, and New Zealand, along with a few nations in Latin America, CF NBS has yet to be implemented in Asia. Other healthcare challenges have taken priority there and uncertainty has prevailed about CF incidence. Yet, the emerging data summarized below suggest a significant incidence in the Indian subcontinent. Consequently, we consider India to be the next frontier for studying CF NBS and, for a variety of reasons described herein, also the *best frontier* for achieving all the benefits of CF NBS.

## 3. Global Challenges and Barriers: *Nothing Worth Having Comes Easy*

### 3.1. Experiences and Lessons Learned in Advising Ten Countries

While having the privilege of serving as the CFF’s National Facilitator for Newborn Screening and Quality Improvement from 2007 to the present, the senior author (PMF) also extended his advisory role to ten other countries and learned about numerous barriers. The first lesson learned was that insufficient leadership at either the NBS laboratory level or among complacent CF physicians has often been limiting, and, without leadership, change can be impossible. All healthcare, like politics, is local, but some public health and clinical programs need to be implemented and managed regionally or nationally, and this broader effort has been challenging. Success in NBS typically requires the engagement of government organizations (bureaucracies) that are usually slow to make decisions and commit resources. When CF NBS was initially proposed in many regions, funding limitations frequently were cited by bureaucrats as barriers, such as in Ireland. Other sources of financial support such as research grants helped many regions, such as in Colorado and Wisconsin. Interestingly, once a government decision is made to screen newborns, adequate fiscal support can suddenly appear. It also became clear in most countries that bureaucratic leaders have limited knowledge of NBS and/or even the distinctive features of infant care. Robert Guthrie overcame this challenge by organizing parent groups that pressured government agencies to advocate for PKU screening [15]. Although parental involvement was helpful for CF in some regions, the veto power of bureaucrats sometimes proved impossible to overcome initially, but, eventually, the rationale and impetus proved compelling and internationalization proceeded throughout the Western world.

### 3.2. Follow-Up Center Issues

Because of the assumption in the USA and elsewhere that the follow-up of positive NBS tests would be readily managed, it was surprising to learn in some regions that CF center organization and capacity were limiting. Just as surprising was the realization that some influential CF physician leaders vigorously opposed NBS even after it had been proven beneficial. Other leaders felt that nutritional benefits were not enough, and that screening should await the demonstration of pulmonary benefits, no matter how long or difficult it would be to show them. In one region, some CF center leaders were resistant because of a bias towards intervention in sick children rather than prevention, pointing out that they became pulmonologists to manage chronic lung disease and “not do well baby care”. This attitude is more common, even today, than one would imagine. But intervention rather than prevention has dominated medical practice philosophy for centuries.

### 3.3. CFTR Genetic Analyses

The evolution of screening algorithms has been an interesting topic during the past three decades. After the IRT/DNA algorithm was proven superior to IRT/IRT by 2004 [5], some labs were resistant to using analytic DNA technology to detect variants in the cystic fibrosis transmembrane regulator (*CFTR*) gene, while a few countries such as Germany and Austria were unable to implement DNA/*CFTR* analyses on babies because of legal constraints. Others, particularly in Latin America, did not have sufficient information on pathogenic *CFTR* variants in their populations, so they often proceeded with the original NBS test, the IRT/IRT algorithm [3,14]. Other countries, however, concluded that more knowledge about the *CFTR* variants in their population was essential before embarking on a NBS program that, ironically, would enable them to satisfy that goal through IRT/IRT plus genetic analyses on those with positive sweat tests. Lastly, one country, Israel, has opted, thus far, to rely on population and prenatal genetic testing as a preventive strategy rather than implement NBS.

## 4. Overview of the Indian Subcontinent (ISC) and Its Unique Features

India is a unique land in almost every aspect imaginable. Its rich history includes the world’s oldest urban developments and architecture, reflecting the brilliance and leadership talents of early settlers. The size of the Indian population and its diversity are unique and astonishing as described below. Its geography and climate have always been challenging but manageable because of the ingenuity of the Indian people in managing resources. As Figure 1 shows, hugely populated cities are distributed throughout the country. Rural areas are also heavily populated but not as densely inhabited. Traditions vary throughout the land and are deeply respected and practiced while blending in with modern activities—a special combination of the very old and new. Currently, India has the world’s fifth largest economy, even though, on a per capita basis, it is classified as a lower–middle-income country (LIMC) and has a high debt level. Projections suggest that India will continue its economic boom and emergence as a global superpower.

There are more than 1.4 billion people living in India, which is 17% of the world‘s population. It now exceeds China as the most populous country. Considering the countless number of ISC inhabitants who have settled elsewhere due to the ongoing “Indian diaspora”, the worldwide impact of Indians is remarkable. Genetic studies [16] show two ancestral populations: Ancestral North Indians (ANI), related to Central Asians, Middle Easterners, Caucasians, and Europeans; and Ancestral South Indians (ASI), not closely related to groups outside the subcontinent. In addition, early genetic mixing with Middle Easterners, Greeks, and Romans was prominent among the south Indian people, reflecting the active trading by the early Indians.

Later, of course, colonization by Portuguese, French, and, especially, British people has influenced the genetic makeup of current Indians. Perhaps even more influential genetically has been the millennia-long tradition of arranged marriages among first cousins, which persists today and has led to a very high level of consanguinity. According to India’s National Family Health Survey-5 (NFHS-5) [17], 11% of marriages in India are consanguineous, and, interestingly, consanguinity rates in South India are reported to be as high as 28%. All these factors are relevant to CF incidence.

## 5. Cystic Fibrosis in ISC Populations

### 5.1. Relevant Observations in the United Kingdom

Observations over many decades in the UK have clearly identified a substantial number of ISC people with CF, including an increasing number of cases diagnosed through NBS since the national program began 2007. In the 2023 UK Registry (https://www.cysticfibrosis.org.uk/sites/default/files/2024-11/CFT_2023_Annual_Data_Report_Oct2024%201.pdf accessed on 15 April 2025), there are 290 ISC patients reported, which shows an increase of 84 during the past decade. Although this likely reflects better recognition through NBS, it is certainly an underestimate because the four-variant DNA/*CFTR* tier of the national UK algorithm will miss the majority of ISC babies. Infants who are often diagnosed through CF NBS are detected because of the very high IRT “safety net” used in the UK. Calculations based on the number of ISC-derived people now living in the UK (3.5 million) allow the extrapolation of the known 290 patients to a minimum CF prevalence of 1:12,000.

### 5.2. Recent CF Data on ISC Inhabitants in India

The genetics of CF in India’s CF population have only been studied to a limited extent, but the data are revealing. Recently, Varkki et al. [18] reported *CFTR* variant data in 120 people with CF. The cohort included 90 samples analyzed by next-generation sequencing at a single center in South India, and 30 others assessed with clinical exome sequencing at other laboratories in India. The results revealed 55 different variants, 49 of which were known and 6 that were novel. They included missense, nonsense, frameshift, and splice variants, along with deep intronic variants and large deletions. The allele frequency of F508del was 27% (64 of 240), in contrast to its common ~70% in patients with Northwestern European ancestry. Importantly, 64 people with CF (53%) had variants that would make them eligible for highly effective modulator therapy, including 55 (46%) people with CF who would be eligible for Elexacaftor/Tezacaftor/Ivacaftor and 20 (17%) who had Ivacaftor-responsive variants.

The median age of death for people with CF in India is 5 years old—a tragic figure when compared to the tenfold-higher median longevity of patients in the USA, which was 61 years, based on the 2023 CFF Registry data [https://www.cff.org/medical-professionals/patient-registry accessed on 15 April 2025]. Undoubtedly, many ISC children with CF die undiagnosed, as increasing evidence suggests that CF-related deaths are underrecognized while masquerading as recurrent pneumonias, malnutrition, and dehydration. In a retrospective study of infant deaths between one and six months of age over three years, 7.4% of deaths were related to CF that was not diagnosed at the time of death [19].

Because of the high mortality of undiagnosed and/or late-treated CF among Indian children, it is not possible to determine an accurate incidence. Indeed, this will only be possible through CF NBS. Thus, estimates have been based on extrapolations from the *CFTR* variant carrier frequency. With a goal towards planning a CF NBS pilot project, a prospective study was carried out to estimate the carrier frequency of *CFTR* variants among 500 neonates born at the same institution in South India. The data revealed a carrier rate of any pathogenic *CFTR* variant of 1:41, with an estimated incidence of CF with two variants at 1:6588 calculated by the Hardy–Weinberg principle. With an annual birth rate of 24 million neonates, this suggests that as many as 3600 infants are born with CF annually in India.

The above estimates were supported recently by a small NBS study [20] in Mysore, Karnataka by a team of geneticists who screened a mostly rural population of 5157 babies and found one CF case. Their project demonstrated the feasibility to screen concurrently for seven genetic conditions at a cost of about $20 U.S. dollars for the entire panel.

Such an incidence, higher than previously expected, should not be surprising. Several studies using NBS data in many states of India have revealed that congenital hypothyroidism occurs with a high frequency—even higher than Western countries [20,21,22,23]. In addition, NBS studies and a recent evaluation of CYP21A2 pathogenic variant carriers have revealed that there is a remarkably elevated risk for congenital adrenal hyperplasia in the ISC population [24,25]. Other studies using NBS have led to the discovery that a variety of inborn errors of metabolism such as hyperphenylalaninemia, tyrosinemia, and maple syrup urine disease (MSUD) are prevalent in the national population and are largely responsible for intellectual disabilities in children [26]. In addition, glucose-6-phosphate deficiency (G6PDD) is prevalent [27].

## 6. India’s Healthcare and Public Health Organization

As in many countries, healthcare in India is delivered by two systems—private and public. Numerous private hospitals and clinics serve the middle class and wealthy people who can afford to pay cash for their care. These are typically corporate-owned facilities that are highly advanced technologically and provide self-pay care comparable to that available in the Western world. They account for much of India’s healthcare and include all the diagnostic services, including excellent clinical laboratories. The government-provided healthcare facilities are also technologically advanced but can be understaffed and less efficient. In contrast to the private sector, they are dependent on government bureaucracies for decision-making about funding for equipment and services. Both systems are committed to a variety of innovations [28] and digitalization [29].

Public health in India is well-organized through a multitiered system involving state and local levels that are under the ultimate control of the central/national leadership, particularly the Ministry of Health and Family Welfare. This organization develops national health policies, programs, and regulations. On the state level, health departments are led by a health minister who is responsible for overseeing core operations such as local hospitals and government health centers. At the district level, through its community programs, both preventive and basic treatment services are ensured through a variety of urban and rural centers. The districts are key to public healthcare delivery. Decisions on the implementation of CF NBS are likely to be made at the district level for government-funded care, while private hospitals may become independently responsible for babies born in their facilities.

## 7. The *Best Frontier*—Potential Uniquely Valuable Impact of CF Newborn Screening in India

There have been many regions of the Western world that have benefited in a variety of important ways when CF NBS was introduced. A good example is France, where pioneering efforts in Normandy during the 1980s ultimately led to a national CF center re-organization and improved care, and, later, similar benefits occurred in Russia. In the USA, the elimination of the common “diagnostic odyssey” [5] has greatly benefited parents, while CF centers have been obliged to improve the efficiency and accuracy of diagnoses as well as communications.

But these benefits pale in comparison to what can be achieved in India, assuming, conservatively a 1:7000–12,000 incidence based on the preliminary data summarized above. Undoubtedly, CF NBS will save the lives of babies there who have been dying undiagnosed—probably in large numbers due to hyponatremic/hypochloremic dehydration, pneumonia, and severe protein-energy malnutrition. Its implementation will greatly advance the current interest in achieving global diversity, equity, and inclusion to address one of the long-term failures of the CF field—the equitable diagnosis of brown and black babies who were traditionally not considered susceptible to this “white person’s disease” [9]. The opportunity for discovery through a combination of service and research in India is unprecedented and unparalleled in both scale and scope. NBS research is undoubtedly needed to establish the incidence in India and, eventually, the presumed regional variations. CF NBS will enable the further expansion of the list of pathogenic *CFTR* variants added to the CFTR2 database [30], which is deficient in Asian *CFTR* variants. NBS will also facilitate the study of unique clinical risks associated with hot weather such as Pseudo–Bartter syndrome [31,32], because they can be studied more thoroughly with a larger number of patients. Because of what we already know about the rare ISC variants affecting CFTR protein production, a new cohort of research subjects will be generated for future clinical trials to help ensure that “no one is left behind”, which the CFF has committed to, as part of its mission.

Moreover, the significant halo effect of CF NBS in India is likely to be substantial in a way that the Western world has not experienced. In fact, CF NBS was introduced in Western civilization countries after more than 20 other genetic conditions were included on the screening panels, but it could be prioritized in NBS panels in regions of India. If the incidence is indeed in the range of 1:6588–12,000, as the data described above indicate, NBS will lead to about 2000–3600 new CF diagnoses annually among the 24 million Indian births/year—a staggering number that would exceed the total number of new diagnoses in North America and Europe combined. Thus, CF history would be rewritten, and new insights could be gained about the elusive selective advantage of *CFTR* variant carriers.

## 8. CF NBS Special Challenges and Strategic Considerations in India

Implementing, and achieving the benefits of CF NBS in India will present the greatest logistical challenge ever faced because of the huge scale, diverse habitation patterns, and very limited previous experience there with any kind of population screening. Its development should be guided by the knowledge gained elsewhere over four decades. Thus, India will need to follow the incremental, sequential strategy used in other large counties and expect an evolving, gradual progress in a timeframe contingent on what is learned over time and the resources that become available. Panchbudhe et al. [26] summarized the main challenges that preclude universal NBS in India as follows: (1) a limited awareness of NBS among medical providers and the public, (2) a significant heterogeneity in population genetics, socioeconomic resources, health care infrastructure, and disease prevalence, (3) funding limitations due to the enormous population size, (4) inadequate pre- and post-test counseling, and (5) difficulties in ensuring the quality control of the process. In addition, ensuring follow-up has proven difficult [20] for many reasons, like changing mobile phone numbers, even though most people have one; cultural issues including potential family disrespect of mothers when children are born with special needs; and government funding capacity. But the populations served by private hospitals should be able to cover the small self-pay amounts for NBS tests, and the relatively low personnel costs for the follow-up component will be advantageous.

Although Therrell et al. [33] predicted that NBS programs would be developed at the state level, we believe that the district level is more likely to be manageable due to the huge populations in the Indian states such as Utter Pradesh with 240 million and Tamil Nadu with 77 million inhabitants—the most populous states in the north and south, respectively. We believe that it would be ideal to initiate CF NBS in large birthing hospitals and expand the population coverage on a hospital-by-hospital basis, and region-by-region basis.

NBS has been performed with cord blood as noted earlier, while heel prick blood collection has not been routinely practiced, particularly in well babies, because there is notable hesitation among new parents due to the discomfort their infants would suffer. But pilot studies of the screening for congenital hypothyroidism have demonstrated that cord blood samples suffice [20] and have many advantages over heel prick sampling (e.g., efficiency and acceptability). Another special challenge that India faces is the considerable number of rare *CFTR* variants that preclude the equitable use of the IRT/DNA algorithm, as was demonstrated in the Mysore, Karnataka NBS project [22], unless next-generation sequencing could be applied affordably to interrogate the entire gene.

The Department of Science and Technology of the Government of India recently supported NBS for five common treatable disorders, while also acknowledging the need for an operational infrastructure to initiate and sustain a mass-scale NBS program [33]. Similarly, the UMMID initiative by the Government of India recommends newer public sector programs targeting genetic disorders, including CF, based on specific genetic risk stratification criteria. These initiatives are appreciated and supported by the Indian Academy of Pediatrics.

According to the recently published [33] Kathmandu Declaration in September 2024, newborn bloodspot screening program representatives from 12 nations in the Asia Pacific region, including India, conferred and emphasized that NBS is an important tool in the prevention of disease, disability, and death in children and, thus, should be a key part of the comprehensive public health system in their countries. Key facets of the Declaration regarding NBS were (1) collaborative educational ventures, (2) an imperative need for population studies, (3) quality control, (4) advocacy to policymakers, and (5) the improvement of regional health infrastructure.

Therefore, there is an imperative need to improve resource allocation for CF NBS by providing more evidence of the disease burden of CF and opportunities for prevention, which our team plans to generate. We believe that this can best be accomplished by sequential plans to initiate institution-based NBS—a strategy that proved successful in the USA and many other countries. Because *every journey begins with a single step*, it is now timely to launch CF NBS in an Indian institution with an established CF program like the Christian Medical College and Hospital of Vellore, Tamil Nadu with its 10,000 annual births [18,19]. From extensive analyses, we believe that the original CF NBS algorithm, IRT/IRT, is the most equitable and cost-effective method. In fact, it has been carried ut at the Christian Medical College and Hospital for $3 per test. Although there will be challenges related to this biomarker’s heat lability and selection of cutoff values, the four decades of experience and recent research data on IRT [34,35,36] will facilitate proceeding with a pilot study.

## 9. Where Is It *Worthwhile* to Screen for CF? India

Scotet et al. [14] published a critical analysis 5 years ago entitled *Newborn Screening for CF across the Globe—Where Is It Worthwhile?* Of the ten criteria they listed, South India meets most of these. Although the “universal collection of dried blood spots specimens”, is not established in South India, cord blood samples could potentially substitute and may be better, as has been shown in the screening for congenital hypothyroidism [21,22]. South India has tertiary centers that offer standardized laboratory services to quickly analyze specimens for IRT and well-functioning CF teams that can assume responsibility for follow-up, diagnosis through sweat testing, genotyping, and care [18,20]. Scotet et al. [14] stated that “the incidence of CF must be high enough to warrant CF care centers in the NBS region”. The data summarized above regarding estimated incidence suggest that this requirement will be met. Research and feasibility projects on NBS will be the key to clarifying this and other issues—just like in the Western world.

## Figures and Tables

**Figure 1 IJNS-11-00027-f001:**
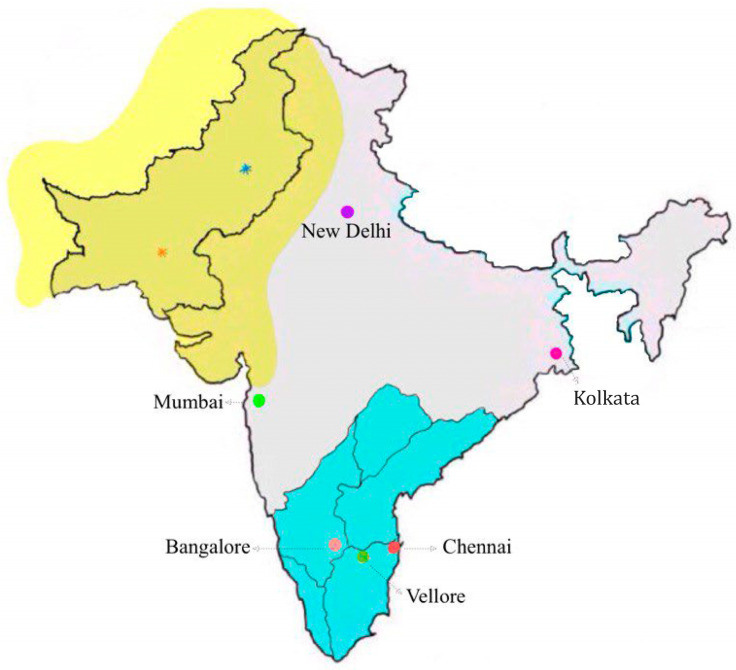
Map of the Indian subcontinent featuring India, showing some of the most populous and best-known urban regions and the five states of South India in blue. The large cities range in populations from 34 million people living in New Delhi to 12 million in Chennai (average = 19.5 million). The map also shows Pakistan and an area in yellow where the world’s oldest civilization thrived in the Indus Valley as determined particularly in the excavated archaeological sites of Harappa (blue dot) and Mohenjo-daro (red dot). The district of Vellore is also identified where the Christian Medical College’s CF program is currently planning to implement CF NBS as a pilot project to study feasibility of screening with cord blood, follow-up, and benefits.

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
