# Peer review of "India: The Last and Best Frontier for Cystic Fibrosis Newborn Screening with Perspectives on Special Challenges"

_2409-515X, 2025, doi:10.3390/ijns11020027_

Round 1
Reviewer 1 Report
Comments and Suggestions for Authors
I thank the Authors for their paper, for the important and precise historical description of the use of neonatal screening for CF in the Western world and for the suggestions they propose for a country like India. I agree that starting with the implementation of neonatal screening for CF in a single district can really be the first step to improve the early diagnosis and treatment of affected subjects, to subsequently obtain all the benefits that cascade from NBS, to then obtain its wider diffusion.
Author Response
I thank the Authors for their paper, for the important and precise historical description of the use of neonatal screening for CF in the Western world and for the suggestions they propose for a country like India. I agree that starting with the implementation of neonatal screening for CF in a single district can really be the first step to improve the early diagnosis and treatment of affected subjects, to subsequently obtain all the benefits that cascade from NBS, to then obtain its wider diffusion.
Response: Thank you for your comments and agreement with our views.
Thank you for your comments and agreement with our views.
Reviewer 2 Report
Comments and Suggestions for Authors
This is an interesting and well written opinion piece. The points are well argued. However, there is an elephant in the room which is the very low level of newborn screening in India for any disorder (including the simple to detect and cheap to treat congenital hypothyroidism). And where there is sample collection and testing, followup of positive tests and treatment outcomes are often lacking. Various authors have suggested reasons for this - practical (eg lots of mobile phones but numbers change often) cultural (eg family disrespect of mothers when children are born with special needs) and of course financial. Please consider adding a paragraph putting CF screening in the context of screening for any other disorder / no screening.
Author Response
This is an interesting and well written opinion piece. The points are well argued. However, there is an elephant in the room which is the very low level of newborn screening in India for any disorder (including the simple to detect and cheap to treat congenital hypothyroidism). And where there is sample collection and testing, followup of positive tests and treatment outcomes are often lacking. Various authors have suggested reasons for this - practical (eg lots of mobile phones but numbers change often) cultural (eg family disrespect of mothers when children are born with special needs) and of course financial. Please consider adding a paragraph putting CF screening in the context of screening for any other disorder / no screening.
Response: Thank you for your comments. We agree about the “elephant in the room” and have added a paragraph to cover both the barriers that are less challenging in the Western World and also “putting CF screening in the context of screening for any other disorder.”
Round 2
Reviewer 2 Report
Comments and Suggestions for Authors
Thank-you for adding a comment about lack of screening in India.